# Oral Ulceration with Bone Sequestration: Key Insights for Clinicians and Their Relevance in Contemporary Clinical Practice—A Narrative Systematic Review

**DOI:** 10.3390/healthcare13111350

**Published:** 2025-06-05

**Authors:** Mariana Agra Monteiro, Lázaro Andrade Rios da Silva, Sebastião Silvério Sousa-Neto, Ricardo Luiz Cavalcanti de Albuquerque-Júnior, Cassiano Francisco Weege Nonaka, Pollianna Muniz Alves, John Lennon Silva Cunha

**Affiliations:** 1Department of Dentistry, State University of Paraíba (UEPB), Campina Grande 58429-500, PB, Brazil; mariana.monteiro@aluno.uepb.edu.br (M.A.M.); cassiano.nonaka@servidor.uepb.edu.br (C.F.W.N.); pmunizalves@servidor.uepb.edu.br (P.M.A.); 2Center of Biological and Health Sciences, Federal University of Western Bahia (UFOB), Barreiras 47810-047, BA, Brazil; lazaro.s3280@ufob.edu.br; 3Department of Oral Diagnosis, Piracicaba Dental School, University of Campinas (UNICAMP), Piracicaba 13414-903, SP, Brazil; s242769@dac.unicamp.br; 4Department of Pathology, Federal University of Santa Catarina (UFSC), Florianópolis 88040-900, SC, Brazil; ricardo.albuquerque@ufsc.br

**Keywords:** oral diseases, oral ulceration, osteonecrosis, oral ulceration with bone sequestration, lingual mandibular sequestration and ulceration

## Abstract

**Background**: Oral ulceration with bone sequestration (OUBS) is an uncommon clinical lesion characterized by painful mucosal ulceration associated with exposed and necrotic bone in the oral cavity, often without known factors inducing osteonecrosis. Despite its potential for diagnostic confusion with more serious lesions, such as medication-related osteonecrosis, OUBS remains poorly understood and underreported. **Objectives**: To systematically review the literature on OUBS and identify its main clinical and radiographic characteristics to assist in early diagnosis and appropriate management. **Methods**: A systematic review was conducted following the Preferred Reporting Items for Systematic Reviews and Meta-Analyses (PRISMA) guidelines. A comprehensive search was conducted in the PubMed, Scopus, and Web of Science databases on 27 January 2024, to identify case reports, case series, and observational studies that described OUBS. Data extraction focused on demographic information, lesion location, clinical features, radiographic findings, possible etiological factors, management, and outcomes. The Joanna Briggs Institute (JBI) critical appraisal tools were utilized to assess the quality of the case reports and series. Data were synthesized narratively due to heterogeneity among studies. **Results**: Fifty-seven patients from 22 studies were included. The male-to-female ratio was 2.5:1, with most cases (71.9%) occurring in males. The mean age was 50.22 ± 11.49 years, with the highest incidence in patients aged 50–59 years (64.9%). Most cases were localized to the mandible (94.7%). Clinically, lesions were shallow ulcers (mean size: 0.75 ± 0.85 cm). Pain was the most common symptom (88.9%). Radiographs revealed no significant maxillary abnormalities, but some cases showed radiopaque areas in the mandible. Surgical interventions were most common (40.4%), with complete healing in 67.7% of cases within 4 weeks. Limitations include the predominance of case reports and series, which limits the strength of evidence. **Conclusions**: Recognizing OUBS can prevent misdiagnosis and unnecessary interventions. Further studies are needed to clarify its etiology and natural history. **Registration**: This review was registered on the PROSPERO database (registration number CRD42024541416).

## 1. Introduction

Oral ulceration with bone sequestration (OUBS) is a condition characterized by the presence of ulcers in the oral cavity accompanied by bone sequestration in the absence of known osteonecrosis-inducing factors [1,2]. A previous study by Lidhar et al. [3] estimated that OUBS accounts for approximately 0.02% of all lesions diagnosed in the service analyzed. Since its initial description by Peters in 1993 [1], a few cases have been reported in the English literature, primarily as single case reports [4,5,6,7,8,9,10,11,12,13,14,15,16] or small series [1,2,3,17,18,19,20,21,22]. These reports, however, often lack crucial details, such as a detailed medical and dental history, information on clinical presentation, anatomical sites, and possible risk factors, necessary to rule out other causes of osteonecrosis. Additionally, various terms have been used to describe this condition, resulting in confusion and a lack of standardization [2].

The predilection of OUBS for some sites, such as the lingual surface of the mandible in the region of the mylohyoid ridge, tori, and exostoses, reflects its probable pathogenic mechanism [2,17,23,24]. These areas of bony prominence are covered by a thin mucosa with less connective tissue and, therefore, are less able to withstand mechanical forces compared to other sites in the oral cavity, increasing the risk of laceration and ulceration [1,10,24]. Secondary inflammation and bacterial colonization compromise the vascular supply of the periosteum, which is already typically reduced in this region, leading to a decrease in pH and altered local oxygen tension, resulting in local ischemia, necrosis, and bone exposure [2]. The presence of bone sequestration maintains the ulcer and impedes the healing process [1,2,18,24,25,26].

The treatment of OUBS does not yet follow a standardized clinical protocol, as it is primarily based on the clinical experience of professionals. Approaches range from conservative procedures, such as the use of topical antiseptics, analgesics, and antibiotics in selected cases, to surgical interventions aimed at removing necrotic bone [2,3,7,8,18,19,21]. The choice of treatment depends on the presence of acute symptoms, and many cases progress with spontaneous expulsion of the bone sequestration [1,11,22].

Although OUBS is considered a rare condition, there is evidence that it is underdiagnosed due to clinicians’ limited understanding, its self-limiting nature, and favorable prognosis, which may contribute to the scarcity of reported cases [2,25]. Since an accurate diagnosis of OUBS is of utmost importance due to its potential to mimic jaw osteonecrosis, a comprehensive understanding of this condition is crucial for improving diagnostic accuracy and guiding effective treatment strategies. Therefore, a systematic review was conducted to establish a basis for a comprehensive understanding of this oral lesion, with a focus on its clinical and imaging characteristics.

## 2. Materials and Methods

This systematic literature review follows the guidelines recommended by PRISMA (Preferred Reporting Items for Systematic Reviews and Meta-Analyses) [27] (Appendix A) and the Cochrane Handbook for Systematic Reviews [28]. In addition, this review was registered on the PROSPERO database under the registration number CRD42024541416.

### 2.1. Data Sources and Search Strategies

A comprehensive literature search was conducted in January 2024, using the electronic databases PubMed/MEDLINE, Scopus, and Web of Science, with no time restrictions. The search strategies, including appropriate keywords and the number of references retrieved from each database, are presented in Table 1. The review question, “*What are the clinical and imaging characteristics of oral ulceration with bone sequestration?*”, was formulated using the PIO framework, which stands for population/problem (patients presenting with OUBS, especially in areas of bone prominence [e.g., tori, exostoses, or the lingual surface of the mandible over the mylohyoid ridge], without any known etiological factors for osteonecrosis), intervention/exposure (presence of OUBS), and outcome (description of clinical characteristics, imaging features, and treatments performed).

### 2.2. Eligibility Criteria and Study Selection

The inclusion criteria encompassed observational studies (cross-sectional, cohort, case series, and case reports) of patients diagnosed with OUBS, published in English, Spanish, French, or Portuguese, containing sufficient clinical and/or histopathological data to confirm the diagnosis of OUBS. Only cases of oral ulcerations accompanied by bone sequestration located in areas of bone prominence, such as tori, exostoses, or on the lingual surface of the mandible over the mylohyoid ridge in patients without any known etiological factors that could induce osteonecrosis were included. Exclusion criteria were studies with unavailable full texts, experimental studies, meeting abstracts, or studies with insufficient diagnostic information.

The selection of studies was conducted independently by two researchers (MAM and LASR) in two distinct stages. In the first phase, duplicated references were removed using a reference management software (EndNote 21^®^, Thomson Reuters, Toronto, ON, Canada), and the titles and abstracts of selected studies were evaluated using the Rayyan^®^ platform [29] “https://rayyan.qcri.org (accessed on 15 February 2024)”. To ensure consistency between reviewers during the selection and quality assessment processes, a calibration exercise was performed using a random sample of 195 articles (approximately 10% of the initial dataset). The inter-rater agreement, measured by the Kappa Test, was 0.916, indicating excellent reproducibility. Those that appeared to meet all inclusion criteria advanced to the second phase, in which the same two authors independently analyzed the full texts to confirm the eligibility criteria. Any disagreement between the two initial reviewers was resolved by a third reviewer (JLSC).

### 2.3. Data Extraction and Analysis

Information such as year of publication, continent and country of origin, number of cases, patients’ sociodemographic data, clinical features, treatments performed, imaging features, duration of the lesion before treatment, recurrence, and follow-up period were extracted from the selected studies (when available). Additionally, information on histopathological aspects and other relevant factors that contribute to understanding this condition was also analyzed.

### 2.4. Quality Assessment

The Joanna Briggs Institute (JBI) critical appraisal tools were utilized to assess the quality of the case reports and series [30,31]. This assessment was independently conducted by three authors (SSSN, MAM, and LASR). To generate the graphs and bias risk summaries for both case reports and case series, the Review Manager (RevMan^®^) [Computer program], Version 5.4, from The Cochrane Collaboration, 2020, was used. The categorization of risk of bias was considered high if the study obtained a response rate of up to 49% ‘yes’, moderate if the study reached 50% to 69% ‘yes’, and low if the study achieved a response rate of at least 70% ‘yes.’

## 3. Results

### 3.1. Study Selection

The study selection process is illustrated in Figure 1. A total of 1953 publications were found in electronic searches. After removing duplicates (n = 346), 1607 articles remained. In the screening phase, only 38 articles were selected for full-text evaluation. Of these, 22 studies [1,2,3,4,5,6,7,8,9,10,11,12,13,14,15,16,17,18,19,20,21,22], reporting 57 cases that met the criteria for OUBS, were included in the present analysis (Figure 1).

A conference abstract, two reviews [23,24], and inaccessible case reports [32,33,34,35,36] were excluded. Additionally, five other cases were excluded for different reasons: the first described bone sequestration occurring in the interdental bone between the upper molars due to the habitual use of a metallic pin [37]. The second was related to an allergic reaction to toothpaste [17]. Both cases occurred in regions devoid of bony prominences and, for this reason, were excluded from further analysis. The third case occurred on the buccal aspect of the mandible, extending from the area of the lower left deciduous canine to the first left primary molar in a healthy 5-year-old boy, with no identifiable local or systemic cause that could explain the spontaneous bone necrosis [38]. Due to the extensive pattern of ulceration, atypical clinical presentation, and location of the lesion in a region devoid of bony prominences, we also chose to exclude this case. Furthermore, we excluded a case in which ulceration followed by bone exposure was associated with pulpal necrosis and represented an osteomyelitis of endodontic nature [39], and a case report of ulceration over a mandibular torus in which microscopic analysis confirmed the vitality of the underlying bone [40].

Additionally, four cases of patients on medications known to induce medication-related osteonecrosis of the jaws (MRONJ) were excluded [20,41,42,43]. One of these cases was described by Bouzouita et al. in 2009 [20] and included in a series of nine cases described as OUBS. The authors described the occurrence of OUBS in a 75-year-old male patient. However, this patient was previously treated with Aredia (pamidronate 90 mg/month) for 12 months for the management of multiple myeloma. For this reason, this specific case was excluded from the analysis, as it represents an example of drug-induced osteonecrosis [20].

Finally, we decided to exclude the report of a patient who presented with bone sequestration two days after receiving a 4 mg intravenous infusion of zoledronic acid for the first time [44]. Although the short time interval makes the occurrence of MRONJ unlikely, we chose to exclude it from the analysis [44].

### 3.2. Study Characteristics

Most papers were single case reports (n = 13; 59.1%) or case series (n = 9; 40.9%) published between 1993 and 2023 that originated from 12 different countries: Canada (n = 12; 21.1%), United Kingdom (n = 9; 15.8%), USA (n = 8; 14.0%), Greece (n = 8; 14.0%), Switzerland (n = 8; 14.0%), Sweden (n = 4; 7.0%), Scotland (n = 3; 5.3%), Brazil (n = 1; 1.8%), Italy (n = 1; 1.8%), Netherlands (n = 1; 1.8%), Turkey (n = 1; 1.8%), and United Arab Emirates (n = 1; 1.8%). Europe (n = 35; 61.4%) and North America (n = 20; 35.1%) concentrated the largest number of cases. The distribution of cases by country is shown in Figure 2. Table 2 provides an overview of the summary data of OUBS reported in the literature (1993–2024).

### 3.3. Clinical Characteristics

Overall, 57 patients from 22 studies were included in the evaluation. Forty-one (71.9%) cases occurred in males and sixteen (28.1%) in females (male-to-female ratio: 2.5:1). The mean age of the affected individuals was 50.22 ± 11.49 years (range: 27–86 years). Patients in the fifth and sixth decades of life were most affected (n = 19, 33.3%, and n = 18, 31.6%, respectively). Regarding the anatomical site, 94.7% of the cases occurred in the mandible (n = 54) involving the mandibular lingual mucosa covering the mylohyoid ridge or on the surface of tori or exostosis. In comparison, only 5.3% affected the maxilla (n = 3).

Clinically, most lesions presented as shallow mucosal ulcers exposing the underlying bone, with sizes ranging from 0.1 to 1.5 cm (mean ± SD: 0.75 ± 0.85). Most patients were symptomatic (n = 52, 91.2%) and reported one or more of the following symptoms: pain (n = 48; 88.9%), soreness (n = 2; 3.7%), discomfort (n = 6; 11.1%), a sensation of a sharp bone spicule in the area (n = 2; 3.7%), a burning sensation when consuming spicy food (n = 2; 3.7%), and/or difficulty chewing (n = 1; 1.9%). The duration of the lesions ranged from 2 days to 1 year, with the majority having a reported evolution time of 2 weeks or less (n = 18; 49.9%), as shown in Table 2.

Periapical, occlusal, panoramic radiographs and/or cone beam computed tomography (CBCT) were performed in some cases (n = 40; 70,17%), and in most of them, no abnormalities were observed in the maxillary bones. However, in some cases (n = 7), occlusal radiographs revealed a localized radiopaque area, indicating the presence of necrotic bone over the lingual cortical plate, or areas of bone rarefaction [1,5,6,17]. Computed tomography (CT) scans were requested in four cases, revealing thinning and/or erosion of the lingual cortical plate of the mandible, possible periosteal reaction, and bone sequestration beneath the lingual cortical bone [8,11,14,15].

Several approaches have been considered in the treatment of oral ulceration with bone sequestration. Most cases were treated using surgical techniques such as sequestrectomy, osteotomy, or surgical debridement (n = 23; 40.4%). However, non-surgical approaches such as waiting for spontaneous exfoliation, non-surgical removal, or debridement of bone sequestra have also been employed (n = 12; 21.1%). It is essential to point out that, in some cases (n = 10; 17.5%), there is no detailed information about the treatment performed, with only “removal of bone sequestration or exposed bone” being mentioned, without specifying whether it was performed through a surgical or non-surgical approach. In 12 cases (21.1%), no direct interventions were performed, and patients underwent exclusively supportive therapies.

In both circumstances, supportive treatment included the prescription of mouthwashes containing substances such as chlorhexidine (0.12 to 0.2%), saline solution, or benzydamine (0.15% *w*/*v*), as well as analgesics and non-steroidal anti-inflammatory drugs, antibiotic therapy, and corticosteroid therapy (Table 2). Other alternative measures, such as the topical application of chlorhexidine gluconate gel (0.2%), sodium hyaluronate gel, and sucralfate [2] or lidocaine ointment 5% [9], have also been prescribed to facilitate the healing process.

Information on the time for complete healing of the lesions was available in only 31 cases (54.4%) and ranged from 2 to 60 days, as detailed in Table 2. Notably, most cases showed complete healing within 4 weeks (n = 21; 67.7%). In three cases, the authors only mentioned that the area entirely healed within a few days, with one of these cases experiencing a recurrence [13]. However, in 26 cases (45.6%), no follow-up information was provided.

### 3.4. Quality Assessment

In total, 13 case reports were incorporated [4,5,6,7,8,9,10,11,12,13,14,15,16]; however, none of them provided positive answers to all eight questions used as a reference. Among the case series evaluated [1,2,3,17,18,19,20,21,22], only one article received positive responses to all 10 proposed questions [3]. However, all studies fulfilled the basic inclusion criteria and were methodologically acceptable for descriptive analysis. Several case reports lacked information about proposed treatment and follow-up. Figure 3 presents the risk of bias assessment for each included study using the Cochrane risk of bias tool. Due to the nature of this review, which includes only case reports and case series, formal statistical assessment of reporting bias (e.g., funnel plot or Egger’s test) was not applicable. The lack of standardized reporting across cases, including incomplete descriptions of treatment and follow-up, further limits the generalizability of findings.

## 4. Discussion

The term “oral ulceration with bone sequestration” (OUBS) has been used in the literature to describe a condition characterized by the presence of oral ulcers accompanied by the exposure of non-vital bone, particularly in areas of bony prominences such as tori, exostoses, or the lingual surface of the mandible without an identifiable cause capable of inducing osteonecrosis [2,3,5]. Few cases have been reported in the literature [1,2,3,4,5,6,7,8,9,10,11,12,13,14,15,16,17,18,19,20,21,22], and existing studies often lack comprehensive clinical information. Although it is usually described as a rare condition, some authors suggest that OUBS is probably underdiagnosed due to several factors, including the lack of knowledge on the part of general practitioners, the presence of mild symptoms that may not encourage the immediate search for medical care, the self-limiting nature of the lesion that favors its spontaneous resolution within a few days, and the possibility of alternative diagnoses [3]. Therefore, it is difficult to determine the true incidence and prevalence of this condition. However, in a previous study conducted by Lidhar et al. [3], OUBS accounted for approximately 0.02% of all lesions diagnosed in the referenced services; however, this prevalence may be underestimated.

Our literature review indicates that OUBS predominantly affects middle-aged adults, with a mean age of 50.2 years (Table 2). This finding is not surprising, since the incidence of oral ulcers tends to increase with advancing age, due to factors such as weakening of the mucosa, immunological compromise, and greater likelihood of tooth loss [45]. The higher incidence reported in males may be related to predisposing factors, such as more intense chewing forces (high chewing load) and the presence of a prominent mylohyoid ridge [3,24]. These factors, combined with the loss of molars, could potentially increase the chances of developing these lesions.

Clinically, OUBS presents as a painful ulcer that varies in intensity, exposing non-vital bone, typically affecting the lingual aspect of the posterior region of the mandible, along the mylohyoid crest, tori, and exostoses [1,2,3]. Upon careful examination of the central area, a hard bony base corresponding to the exposed non-vital bone (bone sequestration) is observed. The non-vital bone may have completely or partially adhered to the area, or in certain circumstances, spontaneous bone sequestration may occur [1,2,17,18,23]. Histopathological analysis shows non-vital bone fragments demonstrating irregular areas of resorption, microbial colonization, and often adherent fragments of granulation tissue [2].

The pathogenesis of OUBS may be associated with recent traumatic events, whether accidental (e.g., excessive forces during chewing, toothbrushing, or bruxism) or iatrogenic (e.g., tooth scaling, dental impressions, traumatic intubation, or tooth extractions [1,4,6,7,9,10,11,12,18,19,23]. In some cases, patients may report a history of aphthous-like ulcers with common precipitating factors such as trauma [17]. Some cases have been reported as “idiopathic” or “spontaneous”, with no identifiable precipitating factors [2,17]. However, the possibility of trauma cannot be definitively ruled out, as injuries occur predominantly in areas of bony prominence susceptible to trauma and lacerations [1,2,6,23].

Lesions are frequently seen in patients who have lost posterior molars or who have received restorations that fail to reproduce the normal lingual inclination of the molars [1,23]. In some cases, occurring on the lingual surface of the mandible, over the mylohyoid crest, the patients lacked permanent molars. The loss of the protective lingual slope over the mylohyoid crest exposes the non-keratinized oral mucosa to prolonged, low-intensity trauma during chewing. Subsequent ulceration leads to interruption of the blood supply, increasing the risk of secondary infection, devitalization of bone tissue, and eventual bone sequestration [1,2,13,22]. Ulcer resolution is hampered by the presence of a non-vital bone fragment and secondary infection in the area [1,2,18].

Mandibular tori and exostoses are secondary areas most affected and prone to ulceration due to tension in the mucosa that covers them. Additionally, the limited amount of connective tissue and its location far from the alveolar blood supply further compromise the healing capacity and defense against infection, potentially leading to bone necrosis [1,2,10,24]. The idea that an aphthous ulcer may be the primary lesion rather than a traumatic ulcer is also valid and may result in the same pathological events [2]. However, some cases occurred in fully dentate patients without unsatisfactory molar restorations. In these situations, although it is difficult to suggest a causal agent, it is important to consider that bony prominences are predisposed to trauma and lacerations [2].

In 2D radiographic examinations (periapical and panoramic radiographs), most cases did not show abnormalities [1,2,3]. However, previous studies have described OUBS as a focal area of bone rarefaction along the lingual cortex of the mandible on mandibular occlusal radiographs [1,3]. Small, ill-defined radiolucencies or interruption of the mandibular or maxillary cortical bone were also observed [1,3]. However, CBCT demonstrated thinning of the cortical surface of the affected bone and signs of bone sequestration, similar to previous studies [11,14,15]. It is essential to note that differentiating MRONJ in the early stages of OUBS can be challenging, as it may present similar radiographic findings. Therefore, it is necessary to obtain a detailed medical history from the patient, particularly regarding the use of medications that can induce bone necrosis.

The differential diagnosis of OUBS can be challenging, and several conditions must be considered in the differential diagnosis, including MRONJ, osteoradionecrosis, congenital or acquired diseases, infectious diseases, and malignancies [2,8,21]. In the case of OUBS, the absence of known systemic factors capable of inducing osteonecrosis [2,24,25,26] and the presence of an often well-defined lesion located in regions of specific bone prominences that heals after removal of the necrotic bone are characteristic findings of OUBS [2,24,25,26].

Although OUBS shares some clinical characteristics with the early stage of MRONJ, such as the preference for the posterior lingual region of the mandible [2] and the possibility of both occurring after a traumatic event or spontaneously without an identifiable precipitating factor, there are notable distinctions. In MRONJ, osteonecrotic lesions are typically poorly defined, associated with various symptoms, and often require systemic administration of antimicrobial agents [2,24,25]. In contrast, in OUBS, the lesions tend to be shallow, well-defined ulcers with mildly erythematous edges, generally healing after the removal of the bone sequestrum [2,24,25]. However, clinicians and dentists lacking experience in diagnosing oral lesions may face challenges in establishing an accurate diagnosis. Hence, enhancing clinicians’ knowledge regarding these injuries is crucial, especially considering the potential for them to mimic the clinical appearance of more serious conditions.

While OUBS has been described for more than three decades [1], there are still no established and definitive clinical protocols for treating these conditions. Likewise, the role of OUBS as a possible trigger for MRONJ is not fully understood. The treatments currently proposed are diverse and based mainly on clinical experience. They range from topical application of chlorhexidine and the use of systemic antibiotics to more invasive procedures, such as debridement and surgical resection of the affected area [1,2,3].

When the exposed necrotic bone is mobile, its removal from the ulcer can be carried out non-surgically, as suggested by several authors [2,3,7,8,18,19,21]. In cases where removal is not feasible, treatment may follow conservative approaches or direct surgical interventions [2]. The choice between these options depends on the presence of acute symptoms [2]. In the conservative approach, the non-vital bone is expected to undergo a spontaneous sequestration process, while the patient receives topical treatment with antiseptics, antibacterials, and analgesics to alleviate symptoms and prevent bacterial colonization [2,6,18,21,23]. Although the use of systemic antibiotics has been adopted in some cases [7,8,12], their effectiveness is questionable due to the reduction of microvascular supply in areas of bony prominences [1,2,6,23]. Therefore, it is prudent to employ these agents with caution. The patient should be monitored until the non-vital bone undergoes spontaneous sequestration [1,11,22] or becomes mobile, allowing for its removal [6,18,22].

On the other hand, advocates of surgical debridement have emphasized that pain and the potential for traumatic ulceration of nearby structures, such as the tongue, by bone fragments, are important considerations that may lead to early intervention [1,2,19]. Mandibular and palatal tori may require surgical removal to avoid potential new trauma and recurrence of the lesions, and loss of lingual inclination will require prosthetic rehabilitation and tooth realignment [1,24].

In addition to the clinical limitations observed in the included cases, it is also necessary to highlight methodological limitations inherent to this systematic review. First, although a comprehensive search was conducted in several databases, it is possible that relevant studies were not included due to language limitations or the lack of indexing in the included databases. Second, all included studies were case reports or series, which have a limited level of evidence and are highly susceptible to publication and selection biases. Although critical appraisal tools from the Joanna Briggs Institute (JBI) were used to measure methodological quality [30,31], it is essential to emphasize that such tools have an exploratory use in this context, since they were developed initially to report more descriptively the quality of reports and not to measure risk of bias with rigor comparable to that of clinical trials. Therefore, the graphs generated on risk of bias should be interpreted with caution.

Moreover, there is a potential for reporting bias, particularly due to the likely underreporting of negative or inconclusive cases, which may overestimate the consistency of clinical features and outcomes described in the literature. The heterogeneity in methodological approaches and clinical data reporting across studies further limits the robustness of pooled interpretations.

These methodological constraints may influence the interpretation of clinical outcomes, such as healing time. Most ulcers healed in less than 2 weeks [1,3,6,7,8,18,19,21]. However, some lesions showed a delay in healing, taking up to 60 days for complete recovery (Table 2). The observed variability in the healing period needs to be interpreted with caution. Determining the precise time required for the complete healing of the lesions is challenging, especially since some patients returned for follow-up after 1 or 2 months from the start of treatment. Although the authors claim complete healing by the end of this period, confirming whether all lesions required this time is difficult. It is recommended that future studies conduct a comprehensive assessment of each case to recognize the influence of individual factors on this healing time. Additionally, the interaction between systemic conditions and the body’s response to specific therapies needs further exploration, as most cases in the literature do not provide accurate information about patients’ systemic conditions or other factors that may significantly impact the recovery process in some cases.

In summary, further studies are needed to determine the true prevalence and incidence of this condition. Furthermore, it is essential to evaluate the proportion of cases that result in significant morbidity, considering their size, duration, and level of pain, as evidenced in the present series. Given that some injuries can be extensive and symptomatic, it is essential to implement additional measures to reduce symptoms and facilitate the healing process, thereby improving the quality of life for affected patients. The creation of a staging system would be particularly beneficial for improving the diagnostic process and therapeutic strategies. Furthermore, it is essential to note that OUBS may play a significant role as a trigger for MRONJ, which warrants further investigation in future studies.

## 5. Conclusions

In conclusion, OUBS remains an underdiagnosed and poorly understood clinical condition, predominantly affecting middle-aged male individuals and occurring in areas of bony prominences in the absence of risk factors capable of inducing osteonecrosis. Clinically, the lesions are characterized by shallow, painful ulcers with exposed necrotic bone, most frequently located in areas of bone prominences. Most cases suggest a traumatic or mechanical etiology despite being often labeled as idiopathic. Lesions usually heal successfully with conservative treatment and surgical intervention, particularly after early removal of any loose necrotic bone tissue. The heterogeneity in treatment approaches and lack of standardized protocols highlight the urgent need for consensus-based clinical guidelines. Future research should prioritize longitudinal and prospective observational studies with standardized diagnostic criteria and outcome reporting to improve our understanding of the natural history, recurrence patterns, and optimal therapeutic strategies for OUBS.

## Figures and Tables

**Figure 1 healthcare-13-01350-f001:**
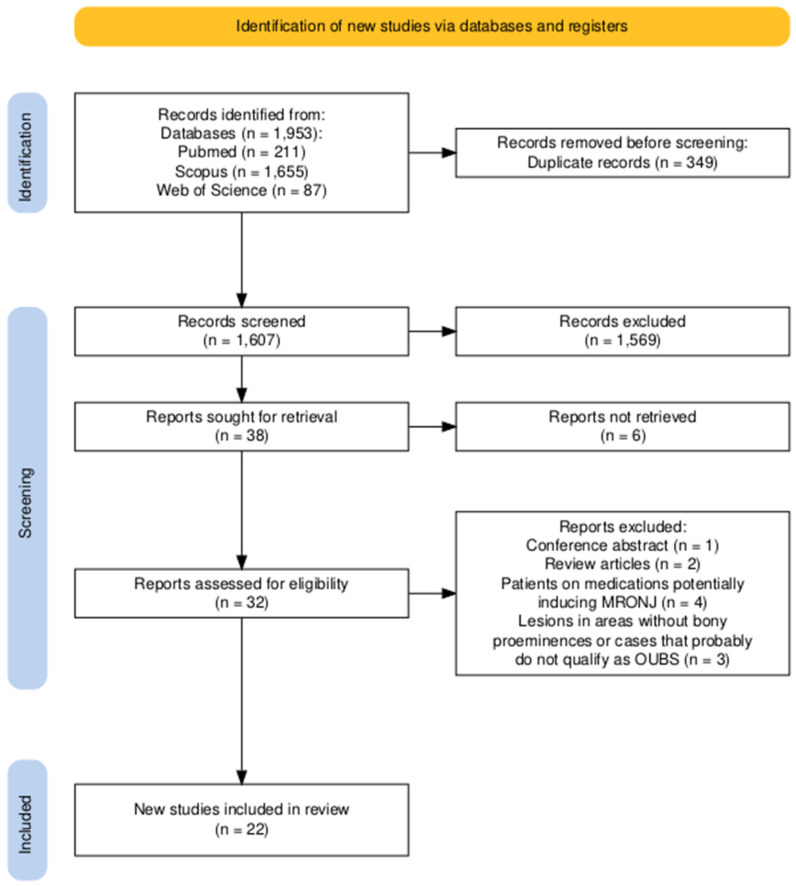
PRISMA (preferred reporting items for systematic reviews and meta-analyses) flow diagram of screening studies for inclusion in the systematic review. OUBS, oral ulceration with bone sequestration; MRONJ, medication-related osteonecrosis of the jaws.

**Figure 2 healthcare-13-01350-f002:**
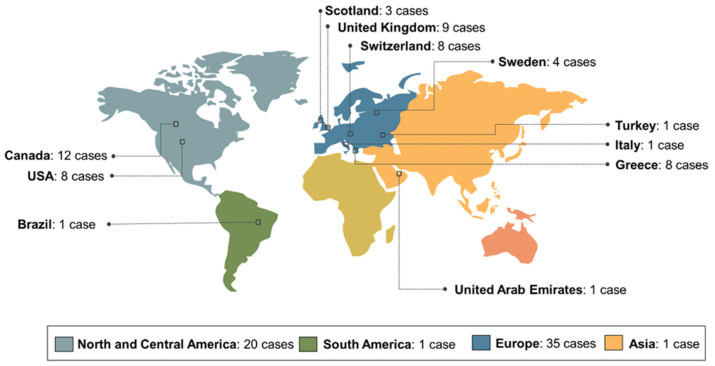
Distribution of cases of oral ulceration with bone sequestration in the world according to the literature review (1993–2024).

**Figure 3 healthcare-13-01350-f003:**
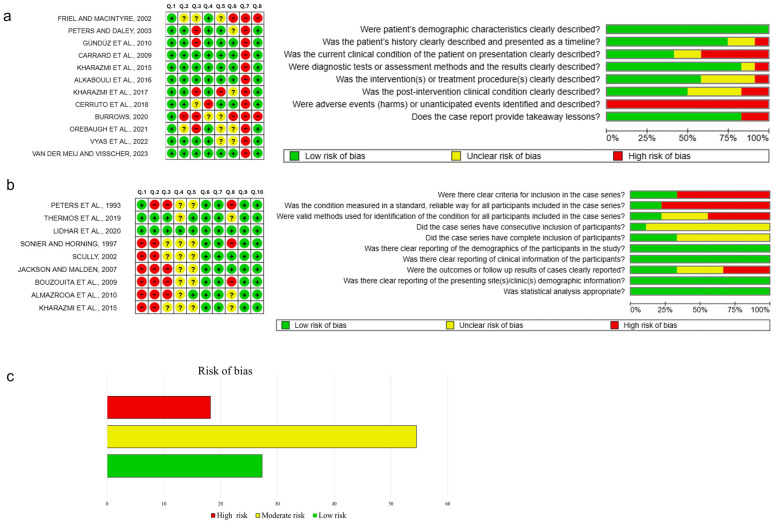
Risk of bias summary and graph for the case reports (**a**) and case series (**b**) using the Cochrane risk of bias tool. The green-colored symbol corresponds to low risk of bias, the yellow symbol corresponds to unclear risk of bias, and the red symbol corresponds to high risk of bias. (**c**) Overall risk of bias was categorized as low when the study achieved ≥70% ‘yes’ ratings, moderate for 50–69%, and high when ≤49% of the criteria were met [1,2,3,4,5,6,8,9,10,11,12,13,14,15,17,18,19,20,21,22].

**Table 1 healthcare-13-01350-t001:** Search strategies with appropriate keywords and the number of references retrieved from each database.

Database	Search Strategy (Search Date: 27 January 2024)	Results
PUBMED	(“Oral ulceration with bone sequestration” OR “Lingual mucosal ulceration with mandibular sequestration” OR “lingual mucosal ulceration with bone necrosis” OR “lingual mandibular bone sequestration” OR “lingual mandibular sequestration and ulceration” OR “lingual mandibular sequestration with ulceration” OR “lingual mandibular osteonecrosis” OR “oral ulceration with mandibular necrosis” OR “mandibular bone exposure and osteonecrosis” OR “bony necrosis and sequestration in the mandible” OR “idiopathic exposure and sequestration of alveolar bone”)	211
SCOPUS	TITLE-ABS-KEY((“Oral ulceration with bone sequestration”) OR (“Lingual mucosal ulceration with mandibular sequestration”) OR (“lingual mucosal ulceration with bone necrosis”) OR (“lingual mandibular bone sequestration”) OR (“lingual mandibular sequestration and ulceration”) OR (“lingual mandibular sequestration with ulceration”) OR (“lingual mandibular osteonecrosis”) OR (“oral ulceration with mandibular necrosis”) OR (“mandibular bone exposure and osteonecrosis”) OR (“bony necrosis and sequestration in the mandible”) OR (“idiopathic exposure and sequestration of alveolar bone”))	1655
WEB OF SCIENCE	“Oral ulceration with bone sequestration” OR “Lingual mucosal ulceration with mandibular sequestration” OR “lingual mucosal ulceration with bone necrosis” OR “lingual mandibular bone sequestration” OR “lingual mandibular sequestration and ulceration” OR “lingual mandibular sequestration with ulceration” OR “lingual mandibular osteonecrosis” OR “oral ulceration with mandibular necrosis” OR “mandibular bone exposure and osteonecrosis” OR “bony necrosis and sequestration in the mandible” OR “idiopathic exposure and sequestration of alveolar bone”	87

**Table 2 healthcare-13-01350-t002:** Clinical and demographic findings, as well as treatment approaches, of OUBS reported in the literature.

Variables	Literature Review (n = 57)	
	n	%
Sex		
Male	41	71.9
Female	16	28.1
M:F ratio	2.5:1	-
Age group		
0 to 9	-	-
10 to 19	-	-
20 to 29	1	1.8
30 to 39	8	14.0
40 to 49	19	33.3
50 to 59	18	31.6
60 to 69	9	15.8
70 to 79	1	1.8
80 to 89	1	1.8
Mean age ± SD	50.2 ± 11.4	-
Range	27–86	-
Location		
Maxilla	3	5.3
Mandible	54	94.7
Ulcer size (cm)		
<1.0	30	78.9
≥1.0	8	21.1
Mean ± SD	0.75 ± 0.85	-
Range	0.1–1.5	-
NI	19	-
Symptomatology		
Pain	48	88.9
Discomfort	6	11.1
Burning sensation	2	3.7
Sensation of sharp bony surface	2	3.7
Soreness	2	3.7
Difficulty in mastication	1	1.9
Symptomatic NS	1	1.9
Asymptomatic	2	3.7
NI	3	-
Duration		
≤2 weeks	18	42.9
>2 ≤ 4 weeks	12	28.6
>4 ≤ 8 weeks	5	11.9
>2 months	4	9.5
Unspecified duration	3	7.1
Range	2 days–1 year	-
NI	15	-
Imaging findings (n = 40; 70.17%)		
Two-dimensional radiographs (n = 36; 90%)		
No alterations	24	60.0
Alteration	12	30.0
Three-dimensional imaging (n = 4; 10%)		
No alterations	0	0
Alteration	4	10.0
Not performed	17	-
Previous traumatic event		
Yes	9	33.3
Does not recall any traumatic event	13	48.1
The authors suggest possible trauma	3	11.1
No	2	7.4
NI	30	-
Treatment		
Surgical (n = 23; 40.3%)		
Surgery/Surgical removal	14	24.6
Sequestrectomy	3	5.3
Osteotomy	4	7.0
Surgical debridement	1	1.8
Bone removed by curettage	1	1.8
Non-surgical (n = 12; 21.1%)		
Spontaneous exfoliation	6	10.5
Non-surgical removal/debridement	5	8.8
Patient removed	1	1.8
Others (n = 22; 38.6%)		
Bone sequestration removal NS	10	17.5
Supportive medications only ^a^	12	21.1
Supportive therapy (n = 48; 84.21%) ^b^		
Systemic antibiotic therapy	13	27.1
NSAID/Analgesic	8	16.7
Corticosteroid	2	4.2
Antiseptic mouthwash ^c^	20	41.7
Chlorhexidine gluconate 0.2% gel or sodium hyaluronate and sucralfate gel	9	18.8
Lidocaine Ointment 5%	1	2.1
Tissucol ^d^	1	2.1
Time required for complete healing (n = 31; 54.4%)		
≤2 weeks	14	45.2
>2 ≤ 4 weeks	7	22.6
>1 month	7	22.6
NS *	3	9.7
Range	2 to 60 days	-
NI	26	-

NI, not informed; NS, not specified; SD, Standard deviation; NSAID, Non-steroidal anti-inflammatory drugs. ^a^ It was not specified whether interventional treatment was adopted for the ulcer (surgical or non-surgical), only the use of supportive therapy being mentioned. ^b^ A combination of different supportive therapies was often employed in the same case. ^c^ Digluconate chlorhexidine (0.2% or 0.12%), saline solution, or benzydamine hydrochloride 0.15% *w*/*v*. ^d^ Topical hemostatic agent/fibrin sealants. * The authors only mentioned that the area had entirely healed within a few days.

## Data Availability

Not applicable. No new data were created.

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
