# Peer review of "Oral Ulceration with Bone Sequestration: Key Insights for Clinicians and Their Relevance in Contemporary Clinical Practice—A Narrative Systematic Review"

_healthcare, 2025, doi:10.3390/healthcare13111350_

Round 1

Reviewer 1 Report

Comments and Suggestions for Authors

The research is sound and very well written.

Some suggestions to be included in the discussion part:

1) The possibilities of genetic background of the OUBS.

2) A susceptibility perspective (differential sensitivity) in patients with OUBS. 

Author Response

Dear Reviewer 1,

We sincerely appreciate the positive comments and suggestions to enrich the discussion.

Genetic background: None of the manuscripts included in this review addressed genetic mechanisms or hereditary predisposition to oral ulceration with bone sequestration (OUBS). For this reason, we did not initially explore this hypothesis. However, we recognize that the investigation of genetic factors related to bone metabolism, inflammatory response and tissue repair may represent a promising line of future research, especially to explain idiopathic or recurrent cases.

Susceptibility perspective: The issue of differential susceptibility has already been indirectly discussed when we highlighted that most cases occur in men, with a predominance in the posterior region of the mandible. We suggest that this may be related to anatomical factors, such as the prominence of the mylohyoid line, which may favor bone exposure in areas subjected to more intense masticatory forces. In any case, we expand on this point in the Discussion, suggesting that individual anatomical characteristics, systemic comorbidities and local conditions may influence the predisposition to the development of the lesion.

Reviewer 2 Report

Comments and Suggestions for Authors

Oral Ulceration with Bone Sequestration: Key Insights for Clinicians and Its Relevance in Contemporary Clinical Practice—A Systematic Review

Reviewer Report

Thank to authors for their study. The authors aimed to systematically review the clinical and radiographic features of oral ulceration with bone sequestration (OUBS) to support early diagnosis and appropriate management. They conducted a comprehensive search of PubMed, Scopus, Embase, and Web of Science databases, analyzing demographic data, clinical and radiological findings, etiological factors, treatment, and outcomes from case reports, series, and observational studies related to OUBS. In total, 57 patients from 22 studies were evaluated, with most cases located in the mandible and predominantly affecting males, especially those aged 50-59 years. They reported no significant abnormalities in the maxilla on radiographic examination, while some cases showed radiopaque areas in the mandible. The characteristic clinical and radiographic features of OUBS were identified, highlighting the importance of these findings for early and differential diagnosis, and concluded that further research is needed to clarify its etiology and natural history.

Abstract:

Abstract has been written in a sufficiently informative manner.

Introduction:

Study provides the relevant background information. Its objective is clearly stated, and the rationale for conducting the study as well as its clinical relevance are explicitly presented. However, it is somewhat brief. Background information should be slightly expanded.

Material and Methods:

Compliance with PRISMA and Cochrane guidelines, PROSPERO registration, detailed search strategies, clearly defined inclusion and exclusion criteria, data selection and extraction process, and quality assessment methods have been specified. Keywords and the databases searched are provided in detail.

However, the following shortcomings need to be addressed:

  • No method for assessing publication bias (e.g., funnel plot or Egger’s test) has been specified. Although the review mainly consists of case reports, evaluating publication bias would still be appropriate.
  • Why were experimental studies not included in the review?
  • It has not been indicated whether there is heterogeneity among the included studies. This is especially important given the methodological variability in case series.

Results:

Selection of studies is illustrated with the PRISMA flow diagram (Figure 1). The number and types of studies, publication years, and geographic distribution are provided in detail. Demographic information of the patients , anatomical location of the lesions, clinical symptoms, and imaging findings are clearly summarized. Appropriate quality assessment tools (Cochrane risk of bias tool) were used for the included case reports and series. As a result, the selection of studies, exclusion criteria, detailed clinical data, and quality assessment were well conducted in the presented findings. However, the lack of follow-up duration and healing data, as well as the absence of publication bias and heterogeneity assessments, require caution when interpreting the results.

Discussion:

Discussion section contains many studies on OUBS in the literature and relates them to the results. However, as a systematic review Discussion, there are some shortcomings: methodological aspects such as the scope of the literature search, the quality of the selected studies, or risk assessment have not been addressed. Including these would further improve this section. The interpretative synthesis drawn from the results should be emphasized more. A more in-depth critique of the methodological quality or reliability of the studies should be added.

Conclusion

Conclusion could be stated even without the need for this review. Please provide a more concrete systematic interpretation and conclusion based on the findings of the articles analyzed.

Author Response

Dear Reviewer 2,

We sincerely appreciate the careful analysis and valuable contributions. Below, we respond point by point to the observations, indicating the modifications made to the manuscript:

1. Abstract:
Reviewer's comment:
Abstract has been written in a sufficiently informative manner.
Response:
We appreciate the positive evaluation.

2. Introduction:
Reviewer's comment:
Study provides the relevant background information… However, it is somewhat brief. Background information should be slightly expanded.
Response:
We have followed the suggestion and expanded the context in the introduction with more information on the incidence/prevalence of OUBS, reinforcing the clinical relevance of the topic.

3. Material and Methods:
Reviewer's comment:
No method for assessing publication bias…
Why were experimental studies not included?
It has not been indicated whether there is heterogeneity…
Answer:
Due to the nature of the review, which included exclusively case reports and series, the application of formal methods for assessing publication bias (such as funnel plots or Egger's tests) was not feasible. We recognize, however, the possibility of reporting bias, especially due to the probable underreporting of negative or inconclusive cases. This limitation was incorporated into the Discussion section of the manuscript.
Regarding heterogeneity, we also included an explanatory paragraph in the Discussion, recognizing the considerable methodological variability of the included studies (regarding design and form of data reporting), which made formal statistical analyses of heterogeneity unfeasible.
In addition, experimental studies were excluded because they did not correspond to the objective of the work, which focused on the clinical and radiographic characterization of real cases described in humans, since OUBS is rarely investigated in experimental models.

4. Results:
Reviewer's comment:
The selection of studies… However, the lack of follow-up duration and healing data, as well as the absence of publication bias and heterogeneity assessments, require caution when interpreting the results.
Answer:
As previously mentioned, the lack of follow-up and outcome data was a relevant limitation observed. We included in the manuscript a critical discussion of this gap in the included studies, emphasizing that this inconsistency compromises a more robust interpretation of clinical prognosis. We recommend that future studies adopt standardized reporting protocols, including longitudinal follow-up and systematic documentation of outcomes. This reflection was included in the final Discussion section, along with considerations on publication bias and heterogeneity.

5. Discussion:
Reviewer's comment:
The discussion lacks methodological considerations… interpretative synthesis should be emphasized more… deeper critique needed.
Answer:
We revised the Discussion section to incorporate a deeper analysis of the methodological quality of the included studies, their limitations, and the impact of this on the strength of the evidence. We also emphasize the interpretative synthesis of the findings, highlighting consistent clinical patterns that can help in early diagnosis, even in the face of methodological limitations.

6. Conclusion:
Reviewer's comment:
Conclusion could be stated even without the need for this review… Please provide a more concrete systematic interpretation…
Response:
We rewrote the Conclusion to more directly reflect the review's findings. We highlighted the recurrent clinical features observed in OUBS cases, their importance for differential diagnosis, and proposed specific recommendations for future standardization of clinical reports. Thus, the conclusion is now more clearly based on the review's systematized results.
We thank you again for your careful reading and for your contributions, which certainly strengthened the quality and clarity of the manuscript.

Reviewer 3 Report

Comments and Suggestions for Authors

Dear Authors,

Thank you for the opportunity to review your manuscript “Oral Ulceration with Bone Sequestration: Key Insights for Clinicians and Its Relevance in Contemporary Clinical Practice—A Systematic Review”. It addresses a relatively rare and interesting condition in dental practice. My comments and suggestions for its improvement are as follows:

  1. The abstract should follow the PRISMA checklist for abstracts. Please follow these guidelines.
  2. The reported incidence of OUBS should be included in the Introduction.
  3. In the sentence ”Since an accurate 61 diagnosis of OUBS is of utmost importance due to its potential to mimic maxillary osteonecrosis…”, do you mean jaw osteonecrosis?
  4. Lines 64-66 - I suggest the authors use passive voice.
  5. Lines 68-70 – the required references are lacking.
  6. The PRISMA checklist should be added as supplementary material.
  7. Please provide the date of registration and the link where the registration protocol can be accessed.
  8. “The review question, "What are the clinical and imaging characteristics of oral ulceration with bone sequestration?" was formulated using the PECOS framework, which stands for population, exposure, comparison (not applicable), outcomes, and study design. “ – The research question is not formulated regarding the PECOS framework. The authors should choose an appropriate PICO modification and strictly follow it to build their question, linking each letter of the abbreviation to the correct part of the question.
  9. The search strategy should include a combination of keywords and MeSH terms. The presented search strategies are not precisely selected, which questions the research’s methodology.
  10. How was the inter-rater reliability evaluated?
  11. “Information such as year of publication, continent and country of origin, number of cases, patients' sociodemographic data, clinical features, treatments performed, imaging features, duration of the lesion before treatment, recurrence, and follow-up period were extracted from the selected studies (when available). ” – This information should be added as a table of the included studies.
  12. The study’s limitations should be discussed at the end of the Discussion.

Author Response

Dear Reviewer 3,

We sincerely appreciate the careful and thorough review provided by the reviewers, which has greatly contributed to improving our manuscript.

  1. Regarding the abstract and PRISMA checklist:

Reply: We have incorporated all important information recommended by the PRISMA checklist for abstracts, taking into account that our review is based on a series of cases.

  1. Regarding the incidence of OUBS in the Introduction:

Reply: We included the only available study reporting the prevalence of OUBS, which is based on data from a single service with a small patient population. We acknowledge that this likely does not reflect the true incidence or prevalence of the condition. Nevertheless, following the reviewer’s suggestion, we have incorporated this information in the Introduction.

  1. Regarding “jaw osteonecrosis” terminology:

Reply: Yes, we meant “jaw osteonecrosis.” We have corrected this throughout the manuscript and highlighted the correction in red.

  1. Regarding use of passive voice in lines 64-66:

Reply: The requested passive voice has been used as suggested.

  1. Regarding missing references in lines 68-70:

Reply: The required reference has been added accordingly.

  1. Regarding PRISMA checklist as supplementary material:

Reply: The PRISMA checklist has been added as supplementary material.

  1. Regarding registration date and link:

Reply: The registration date has been added to the manuscript. The link to the PROSPERO registration protocol is not usually provided in systematic reviews, as it can be easily accessed by searching the PROSPERO database using the registration number mentioned in the manuscript.

  1. Regarding research question formulation and PECOS framework:

Reply: We thank the reviewer for this careful observation. We have corrected this by adopting the appropriate PICO framework and strictly linking each letter to the corresponding component of our research question, as now detailed in the manuscript.

  1. Regarding search strategy and MeSH terms:

Reply: We acknowledge the importance of using both keywords and MeSH terms to ensure comprehensive literature retrieval, particularly in databases like PubMed. In our review, a broad and sensitive search strategy was employed, using multiple relevant keywords and synonyms related to oral ulceration with bone sequestration. This approach aimed to capture all potentially relevant studies, considering that specific MeSH terms for this rare and poorly defined condition are limited or not standardized.

Moreover, given the rarity and heterogeneity of the condition under study, many relevant articles may not be consistently indexed under a unique MeSH term. Therefore, to avoid missing studies, we prioritized an inclusive keyword-based search combined with logical OR operators to cover all plausible terminologies reported in the literature.

Nevertheless, we recognize that integrating MeSH terms could further refine the search, and this will be considered in future updates of this systematic review.

  1. Regarding inter-rater reliability evaluation:

Reply: Inter-rater reliability was assessed using the kappa statistic. This information has now been added to the Methods section. We appreciate the reviewer’s helpful suggestion.

  1. Regarding presentation of extracted data as a table:

Reply: Data such as continent, country, and number of cases are presented in the map created for this purpose (Figure 1). Most other information is summarized in Table 2. The year of publication is detailed in Figure 3, which also shows quality assessment and risk of bias. To avoid redundancy, we opted to maintain this organization.

  1. Regarding dthe iscussion of study limitations:

Reply: Some limitations were already addressed in the last paragraph of the Discussion. However, following the reviewer’s suggestion, we have added a new paragraph explicitly highlighting the study limitations.

Reviewer 4 Report

Comments and Suggestions for Authors

This systematic review is a valuable contribution to oral medicine, offering key insights into OUBS. The study adheres to PRISMA guidelines, employs robust search strategies, and critically evaluates case reports and case series. Few recommendations:

  1. Limitations appear to be missing from the Discussion. Given that most studies were case reports or small case series, limiting generalizability, it is recommended to acknowledge this as a limitation and suggest future multicenter cohort studies.
  2. May consider a scoping review or narrative synthesis if meta-analysis is unattainable.
  3. Attach (upload) any relevant Supplimentary files, e.g. Prisma checklist.

This review will be a useful resource for clinicians managing OUBS.

Best regards,

Your Peer-Reviewer

Author Response

Dear Reviewer 4,

We sincerely thank the reviewer for the careful and constructive evaluation of our manuscript.

  1. We agree that acknowledging the study limitations is important. Accordingly, we have added a dedicated paragraph in the Discussion explicitly addressing the limitations related to the predominance of case reports and small case series, which limit the generalizability of the findings. We also suggest that future multicenter cohort studies would be valuable to better understand OUBS.

  2. We appreciate the suggestion regarding scoping reviews or narrative syntheses. As our review focuses exclusively on a case series of a rare condition, performing a meta-analysis was not feasible due to the heterogeneity and nature of the data. We clarified this point in the manuscript.

  3. The PRISMA checklist has been included as supplementary material as requested.

We believe these revisions strengthen the manuscript and appreciate the reviewer’s helpful comments.

Best regards

Round 2

Reviewer 2 Report

Comments and Suggestions for Authors

Thank to the authors for their revisions. They have given due attention to all the requested revisions. The methodological and conceptual limitations mentioned have been addressed in Discussion section.  As a minor suggestion, I recommend additionally expanding the Introduction section a little more and revising the references to ensure they align with the journal's guidelines.

Comments on the Quality of English Language

English is fine and does not require any improvement.

Author Response

Thank you for the careful and constructive evaluation.

We have expanded the Introduction section by adding a paragraph about the treatment of OUBS. With this addition, we believe the Introduction now provides a concise overview of the main points relevant to the topic.

Reviewer 3 Report

Comments and Suggestions for Authors

I suggest the authors use the PIO framework, as the comparison was stated as "not applicable".

Author Response

Thank you for your suggestion.

We have revised the manuscript to incorporate the PIO framework and indicated the red changes within the text.